# Classification of the trajectory of changes in food intake in special nursing home for oldest-old in the 6 months before death: A secondary analysis

**Sakiko Fukui**[1]*, **Kasumi Ikuta**[1], **Tatsuhiko Anzai**[2], **Kunihiko Takahashi**[2]

**1** Department of Home Health and Palliative Care Nursing, Graduate School of Health Care Sciences, Institute of Science Tokyo, Bunkyo-Ku, Tokyo, Japan, **2** Department of Biostatistics, M&D Data Science Center, Institute of Integrated Research, Institute of Science Tokyo, Bunkyo-ku, Tokyo, Japan

* fukuisakiko.chn@tmd.ac.jp

## Abstract

### Background

For the oldest-old residents around their 90s living in facilities, quality end-of-life care is crucial. While an association between reduced food intake and death is known, specific patterns of intake changes before death are not well-documented.

### Aims

This study aims to classify food intake changes among residents in Japan's special nursing homes during the 6 months before death, enabling precision care for each group using routinely recorded data.

### Methods

Sixty-nine deceased older adults from five special nursing homes were studied over 3.5 years (January 2016 to June 2020). Criteria included: at least six months' residency before death, ability to eat orally during the study period, and death within the facility. We created a time-series dataset for 69 participants, documenting their average weekly food intake (on a scale of 0-10). Subsequently, we used cluster analysis to identify clusters of change in the average weekly food intake from the 6 months before death.

### Results

Eligible residents' mean age was 89.7 ± 6.7 years, and 79.7% were female. Cluster analysis classified 4 clusters of decline in food intake changes during the last 6 months before death: immediate decrease (n=14); decrease from 1 month before death (n=24); decrease from 3 months before death (n=7); and gradual decrease for 6 months before death (n=24).

**Data availability statement:** The data cannot be publicly available as it contains personal information. Corresponding author SF or Osaka University Clinical Research Review Committee (rinri@hp-crc.med.osaka-u.ac.jp) should be contacted if someone wants to request the data.

**Funding:** This study was funded by Grant-in-Aid for Scientific Research B from Japan Society for the Promotion of Science (18H03112). The funders had no role in study design, data collection and analysis, decision to publish, or preparation of the manuscript.

**Competing interests:** The authors have declared that no competing interests exist.

## Conclusion

This study identified four groups of food intake prior to death. Recognizing food intake clusters in practical settings can help manage and provide appropriate end-of-life care in facilities with few medical providers but many care providers.

## Introduction

In 2023, older adults aged 65 years and above accounted for 29.0% of Japan's population [1], which is expected to rise to one-third of the population by 2036 [2]. The death rate in Japan is projected to peak at 1.679 million in 2040 [2]. The advent of a super-aging society with a rising death rate has resulted in a burgeoning interest in end-of-life. The key issue is to improve the "quality of death (QOD)," which is the maintenance of dignity until the end of life [3]. Components of QOD have been found to include "dying in a favorite place" and "maintaining hope and pleasure" [4].

In order to respond to the super-aging society and to react the mulfh-level death in Japan, there is a growing need for facilities such as special nursing homes for older adults. The Japanese government has provided policy guidance with the establishment in 2006 of an "end-of-life care supplement" at special nursing homes for older adults [5]. Special nursing home for older adults is characterized by a lack of medical providers. There is no obligation to assign a full-time physician; three nurses are assigned for every 100 residents [6]. This suggests that end-of-life care in the special nursing home is mainly provided by care providers [7] revealed that if it were possible to recognize the probability of death, it might have enabled residents to die more peacefully in familiar surroundings and with people they know. Therefore, even if the medical system is not well organized, care providers who have limited medical knowledge due to their backgrounds can understand the trajectory of older people to death. This can be based on changes in food intake from routinely recorded real-world data in practical settings. Noting this allows for appropriate end-of-life discussions with medical providers and the provision of care in special nursing homes.

Recently, an association has been observed between declining food intake and death in older adults [8–11]. However, changes in food intake before death are not consistent. Trends in classifying clusters of changes in food intake focusing on the duration of the last half-year before death among older people in the facilities could be identified from data routinely recorded in electronic care files by care providers. This might allow health care providers to deliver timely and appropriate end-of-life care in collaboration with medical providers. Furthermore, we believe that if appropriate end-of-life care is provided based on clusters of food intake up until death, older residents will be able to spend their final moments in their favorite place, maintaining hope and pleasure.

Regarding previous studies that have investigated factors associated with mortality in the older people, there is a wide range of prevalence of low body mass index, malnutrition, and eating disability reported among nursing home residents [12,13]. Another interview study showed that early signs of imminent death in residents included the following: declining participation in social activities; less zest for life, giving up; disinterest in or reduced intake of food and/or fluids [14]. Other studies have also identified an association between reduced appetite and death [15–19], but findings stating the associations between appetite and trajectory to death may not be uniform. In addition, studies clarified a variety of clusters of change in food intake prior to death considering the presence or absence of cancer [20,21], as well as the pre-death symptoms and physical function declines to death [22–26]. However, as far as we know, no studies have evaluated and classified the trajectory of change in food intake before death for older people in facilities.

Therefore, we focused on the daily recorded food intake changes observed by care providers and aimed to classify the trajectory of these changes in the oldest-old, around their 90s, in special nursing homes during the 6 months before death.

## Methods

### Study design, participants, and setting

This study was a secondary analysis of a retrospective cohort study reported in 2024 [27]. We included eligible 69 older adults aged 65 years and above who were admitted to five special nursing homes for older adults in Japan between January 1, 2016, and June 23, 2020. The inclusion criteria in this study were being 65 years of age or older; stayed in a special nursing home more than 6 months prior to death; were able to take oral intake during the study period; and died in a nursing home. The exclusion criteria in this study were deaths outside of the special nursing home for older adults, unable to oral intake during the study period, and missing data at two or more time points during the follow-up analysis (24 points: 6 months before death). We accessed these data on March 25, 2021. We had not access to information that could identify individual particioants during and after data collection.

The participating nursing homes were managed by three organization where located in three regions of the country in Japan. There are three types of nursing homes paid for by long-term care insurance as follows: special nursing homes for older adults, long-term care health facilities for older adults, and long-term care hospitals. A special nursing home for older adults is a type of nursing home that can be classified under Japan's long-term care insurance system. These facilities provide nursing care, including bathing, toileting, meals, other daily living activities, functional training, health management, and medical care. The older adults must be officially recognized as requiring level 3 or higher care need within the range of level 1-7 (level 1–2 support need and level 1–5 care need) under the long-term care insurance system to admit to these facilities [5]. The clinical picture of a person requiring level 3 care need is an older adult who has difficulty getting up, moving, and getting in and out of vehicles on their own and understands the daily schedule. The older adults requiring level 5 care need to have the previously mentioned functioning in addition to dysphagia, disorientation, limited joint range of motion, and motor paralysis [5].

### Data collection

The data for this study were obtained from the electronic care files routinely collected at the targeted special nursing home. The electronic medical files included the date of birth, gender, level of care need, date of admission, date of death, and amount of daily food intake for each older adult. As for the level of care needed, data were used from the earliest time point during the individual follow-up period of 6 months before death. Regarding the assessment of food intake at each meal, it undergoes an inter-rater evaluation process by multiple staff members, and monthly conferences are held to validate the evaluation of food intake data.

Older adults who participated in this study were assigned an identification number and were anonymized for analysis. A list of resident name/identification number combinations was kept at each special nursing home, and careful attention was paid to the protection of personal information. At the targeted special nursing home for older adults, three meals (breakfast, lunch, and dinner) were prepared and served daily. If the residents experienced difficulty eating or swallowing, they were supported by care providers and nurses. The food intake was visually recorded by the care providers who served the meal. The food intake was evaluated for each staple (e.g., rice) and each side meal (e.g., side dish) using a numerical value from 1 to 10, with 10 being the total amount served.

## Data analysis

Regarding the daily food intake, which is the sum of the staple and side meals divided by 2, the average food intake per week for 6 months (24 weeks) retrospectively from death was first calculated.

Hierarchical cluster analysis using Ward's method and squared Euclidean distance as the similarity measure [28,29] was conducted to identify clusters with high homogeneity related to changes in average weekly dietary food intake from 6 months prior to death. Missing data were supplemented with the last observation carried forward (LOCF) method. The analyses were conducted with the number of clusters set at 2 to 6, respectively, and the final cluster model was selected based on the classification of the data and its clinical interpretability.

To interpret the profile of the clusters, we analyzed them in a linear mixed model and graphically illustrated the changes in food intake. For the linear mixed model for each cluster, number of weeks before death, sex, age, and level of care needed were set as fixed effects, and participant ID was set as random effect. Additionally, the characteristics of each cluster were compared using Fisher's exact test and the Kruskal–Wallis test. The Kruskal-Wallis test was used to compare food intake between clusters at each time point. The significance level was set at $P < 0.05$. All analyses were conducted using Python version 3.8.1 for MAC (Python Software Foundation) and R statistical software, version 4.3.0, (R Core Team).

## Ethical considerations

This study was conducted in accordance with the principles of the Declaration of Helsinki and after obtaining approval from the human research ethics committee of the authors' affiliate university (Approval Number: 17411-11). The study adopted opt-out, in which posters were posted at the target special nursing home for older adults regarding the purpose and implementation of the study, instead of informed consent. They provide contact information on the poster and assure the participants that they will not be disadvantaged if they refuse the study and can refuse at any time.

## Results

### Participants characteristics

The eligible participants were 69 of the total 303 who had died by June 23, 2020 (Fig 1). The mean age at death of the 69 participants was 89.7 ± 6.7 years, and 55 (79.7%) were female, and the length of stay was 668.8 ± 357.1 days (Table 1).

A total of 234 participants were excluded (died outside of a special nursing home for older adults (n = 72), stayed in a special nursing home for less than 6 months prior to death (n = 107), were not taking oral intake (n = 51), and were missing data at two or more of the 24 time points (6 months before death) (n = 3) (Fig 1).

### Classification of changes in food intake for 6 months prior to death (cluster analysis)

Of the 25 time points (including the time of death: time 0) for 69 participants (total 1,725 data points), 27 time points (n = 17) had missing data, where each had 1 to 2 missing time points. The most missing were observed at the time of death (time point 0) for 10 participants and at time point 1 for 9 participants.

The means pre-death food intake of the whole participants were 7.58 ± 2.20, 7.27 ± 2.37, 5.40 ± 3.31, 4.51 ± 3.73 for 24 weeks (6 months), 12 weeks (3 months), 1 week before death, and just before death, respectively (Table 2). Individual changes in food intake during the first

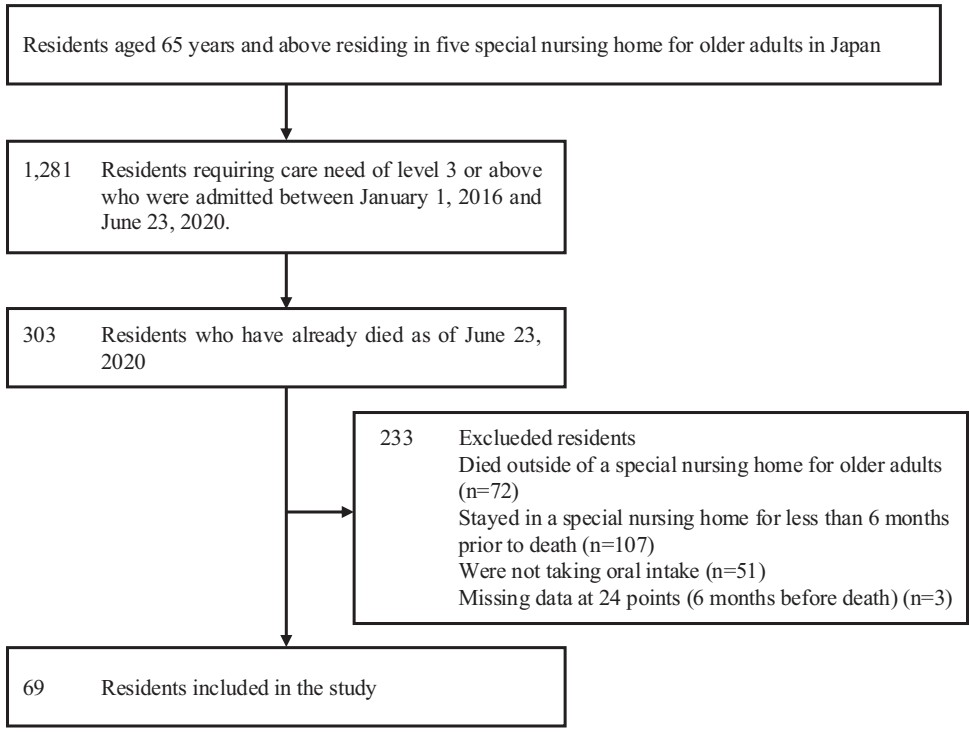

**Fig 1. A flow diagram of the study entry.**

half-year of death showed various clusters of decline. The changes in food intake were classified using cluster analysis into four groups.

The clusters appeared by the analysis with the number of clusters from 2 to 6 are shown in S1 Appendix. We selected the number of clusters 4 as the final model based on clinical interpretability. The profiles of the 4 clusters, confirmed by linear mixed analysis considering covariates, are shown in Fig 2: Group 1 was named "Immediate decrease (n=14)", Group 2 "Decrease from 1 month before death (n=24)", Group 3 "Decrease from 3 months before death (n=7)", and Group 4 "Gradual decrease for 6 months before death (n=24)".

The characteristics of each group are shown in Table 1. Group 2 tended with a longer length of stay and an older age at death. Group 3 tended with a younger age at death and a higher level of care. Group 4 tended with females. However, there were no statistically significant differences in the characteristics of the cluster, except for the facilities

Table 2 is presented with respect to the change in the mean food intake prior to death in each cluster. The Kruskal-Wallis test was performed and significant differences between clusters were found at all time points.

## Discussion

We classified the 4-type trajectory of change in food intake among the oldest-old residents around 90 years living in Japanese special nursing homes during the 6 months prior to death using cluster analysis. Cluster analysis can be used to apprehend individual characteristics not captured by averages of the overall variable by assessing the trajectory of change in food intake prior to death [28]. Thereby, changes in food intake up to end-of-life care can be monitored for each individual. Cluster analysis clusters of change in food intake prior to death

**Table 1. Participants characteristics.**

|  | All | Group 1 (Immediate decrease) | Group 2 (Decrease from 1 month before death) | Group 3 (Decrease from 3 months before death) | Group 4 (Gradual decrease) |  |
|---|---|---|---|---|---|---|
| N (%) | 69 (100.0) | 14 (20.3) | 24 (34.8) | 7 (10.1) | 24 (34.8) | p-value |
| Sex* |  |  |  |  |  | 0.373 |
| Female | 55 (79.7) | 9 (64.3) | 19 (79.2) | 6 (85.7) | 21 (87.5) |  |
| Male | 14 (20.3) | 5 (35.7) | 5 (20.8) | 1 (14.3) | 3 (12.5) |  |
| Age at death* | 89.74 ± 6.66 | 90.07 ± 6.60 | 90.67 ± 6.49 | 87.00 ± 6.30 | 89.42 ± 7.13 | 0.638 |
| Level of care needed* |  |  |  |  |  | 0.475 |
| care need level 3 | 10 (14.5) | 1 (7.1) | 6 (25.0) | 1 (14.3) | 2 (8.3) |  |
| care need level 4 | 19 (27.5) | 6 (42.9) | 5 (20.8) | 1 (14.3) | 7 (29.2) |  |
| care need level 5 | 40 (58.0) | 7 (50.0) | 13 (54.2) | 5 (71.4) | 15 (62.5) |  |
| Admission Period (days)* | 668.77 ± 357.07 | 631.36 ± 380.70 | 709.71 ± 349.04 | 663.71 ± 392.43 | 651.12 ± 360.69 | 0.916 |
| Facility number* |  |  |  |  |  | 0.022 |
| Facility number 1 | 26 (37.7) | 3 (21.4) | 7 (29.2) | 3 (42.9) | 13 (54.2) |  |
| Facility number 2 | 8 (11.6) | 0 (0.0) | 1 (4.2) | 1 (14.3) | 6 (25.0) |  |
| Facility number 3 | 17 (24.6) | 8 (57.1) | 7 (29.2) | 1 (14.3) | 1 (4.2) |  |
| Facility number 4 | 12 (17.4) | 2 (14.3) | 7 (29.2) | 1 (14.3) | 2 (8.3) |  |
| Facility number 5 | 6 (8.7) | 1 (7.1) | 2 (8.3) | 1 (14.3) | 2 (8.3) |  |

[a] Fisher's exact test.

[b] Kruskal–Wallis test.

* N (%) or mean ± SD.

**Table 2. The means pre-death food intake of all participants and each 4 groups for 24 weeks to death (Range 1-10).**

|  | n | 24w | 23w | 22w | 21w | 20w | 19w | 18w | 17w | 16w | 15w | 14w | 13w | 12w | 11w | 10w | 9w | 8w | 7w | 6w | 5w | 4w | 3w | 2w | 1w | 0w |
|---|---|---|---|---|---|---|---|---|---|---|---|---|---|---|---|---|---|---|---|---|---|---|---|---|---|---|
| All | 69 | 7.58 | 7.50 | 7.52 | 7.69 | 7.69 | 7.70 | 7.66 | 7.64 | 7.49 | 7.33 | 7.34 | 7.37 | 7.27 | 7.33 | 7.13 | 7.04 | 6.94 | 6.89 | 6.72 | 6.58 | 6.18 | 5.92 | 5.72 | 5.40 | 4.51 |
| Group1 | 14 | 9.71 | 9.69 | 9.71 | 9.77 | 9.77 | 9.82 | 9.88 | 9.78 | 9.76 | 9.57 | 9.65 | 9.71 | 9.66 | 9.82 | 9.82 | 9.68 | 9.73 | 9.77 | 9.66 | 9.65 | 9.61 | 9.48 | 9.52 | 9.39 | 9.37 |
| Group2 | 24 | 8.37 | 8.17 | 8.40 | 8.74 | 8.71 | 8.63 | 8.48 | 8.51 | 8.26 | 8.24 | 8.28 | 8.43 | 8.45 | 8.48 | 8.09 | 8.42 | 8.28 | 8.47 | 8.49 | 8.42 | 8.13 | 7.51 | 7.20 | 7.02 | 5.90 |
| Group3 | 7 | 8.39 | 8.77 | 8.65 | 8.67 | 8.85 | 8.62 | 8.54 | 8.72 | 8.59 | 8.50 | 8.41 | 7.87 | 7.82 | 8.18 | 7.36 | 6.62 | 6.60 | 5.86 | 4.41 | 3.51 | 2.38 | 2.56 | 2.68 | 2.06 | 0.91 |
| Group4 | 24 | 5.30 | 5.18 | 5.05 | 5.14 | 5.11 | 5.25 | 5.30 | 5.22 | 5.08 | 4.77 | 4.76 | 4.80 | 4.52 | 4.47 | 4.55 | 4.25 | 4.08 | 3.93 | 3.91 | 3.84 | 3.35 | 3.25 | 2.93 | 2.41 | 1.33 |

*The Kruskal-Wallis's test was performed and significant differences between clusters were found at all time points.

were classified similarly to the previous study [30], which resulted from pre-death symptom changes without distinction between patients with and without cancer. This study showed that even among stable older adults without active treatment, changes in food intake prior to death were not uniform and revealed 4-type groups of change.

Special nursing homes for older adults surveyed in this study was characterized as facilities with few medical specialists. We think that the special nursing homes could be considered to have older adults with dementia or who were frail in the study [30]. Due to this characteristics, the facilities do not have a comprehensive understanding of the names of illnesses and medical information, even for residents with medical needs, and this information could not be obtained during this survey. Therefore, we have provided an overview and discussion of each group of change regarding situations other than illness. We believe that the recognition of various trajectories may prompt care providers to consider end-of-life decline, aiding

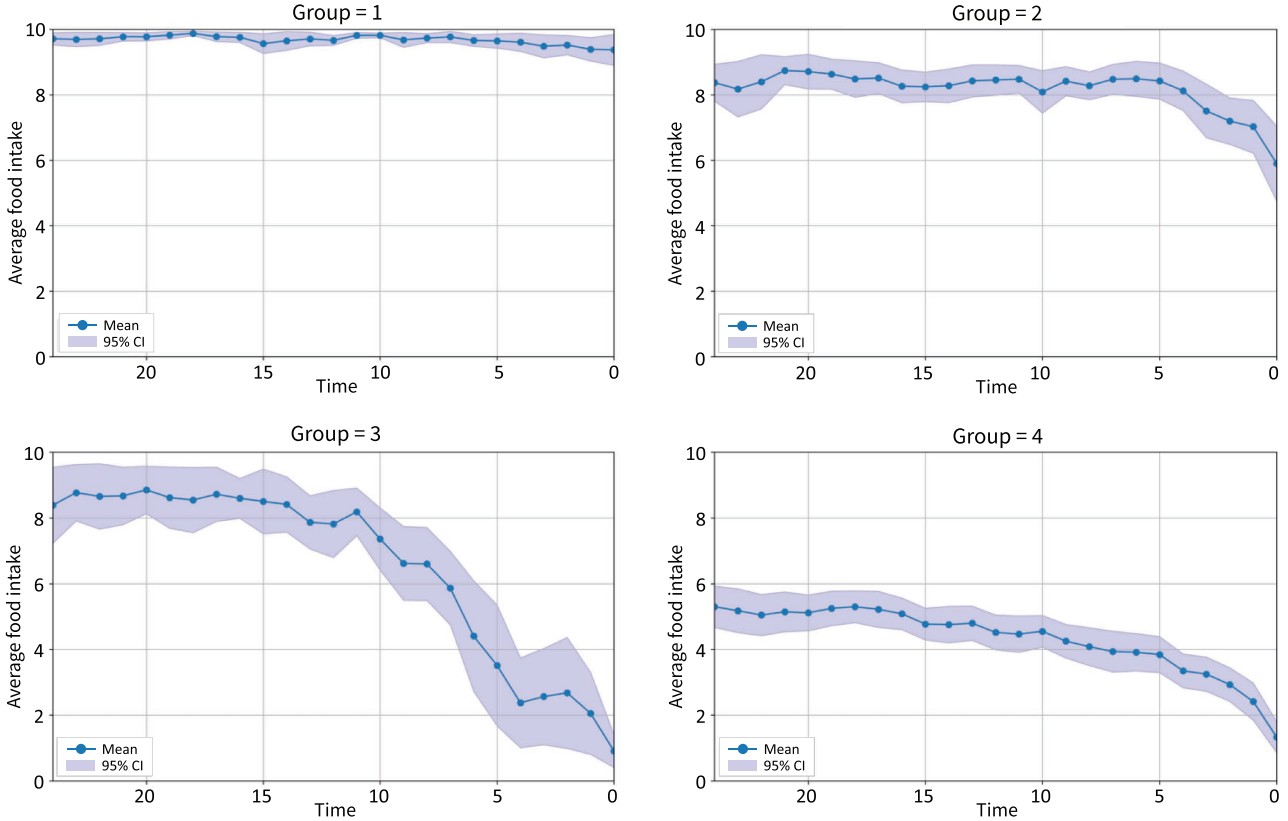

**Fig 2. Classification of changes in food intake (vertical axis range 1-10) at 6 months prior to death (cluster analysis).**

care management and expectations, in collaboration with medical providers such as nurses. Therefore, we have provided an overview and discussion of each group of change regarding situations other than illness.

Group 1 experienced decreased food intake immediately prior to death and possible sudden death. Sudden death causes include acute myocardial infarction, stroke, and dissecting aortic aneurysm, with the majority being acute myocardial infarction among older adults [31]. In Japan, acute myocardial infarction accounted for 30,578 (2.1%) of the 1,439,856 all-cause deaths, stroke for 10,4595 (7.3%), and dissecting aortic aneurysm for 19,351 (1.0%) according to the 2021 Vital Statistics. Older adults accounted for 1,314,242 (91.3%) of all deaths, 26,874 (87.9%) for sexual myocardial infarction, 95,900 (91.7%) for stroke, and 17,538 (90.6%) for dissecting aortic aneurysm [32]. Even if an older adult is residing in a special nursing home, there is still a certain amount of potential for sudden death. To achieve the desired end-of-life care for older adults, discussions regarding end-of-life choices should be performed as early as possible, should include their family, care and medical providers, and be based on the assessment of the timing of death from patterns of food intake changes.

Group 2 in this study experienced decreased food intake 1 month prior to death and may have cancer. According to a previous study [33], approximately 26.0% of nursing home residents have cancer. Nursing home residents do not have regular checkups; therefore, cancer may be slowly growing without showing symptoms of cancer [34]. Although nursing home residents do not have many thorough medical examinations (computed tomography,

gastroscopy, and colonoscopy) at the end of life, they may have a cancerous growth that is not detected until after they have been diagnosed [35]. When they are not feeling well and tests are conducted, it is observed that it may be due to cancer [35]. Thus, there is a possibility that some older adults may have cancer in their bodies that is not detected until they die. In the case of older adults, the decline in physical and cognitive functions due to increasing age should be considered, and appropriate treatment should be provided to each individual, including the option of not choosing to treat. Care providers should consider the cluster of food intake until death, the resident's perspective on life, and the wishes of the family to support residents in living their own lives and spending as much time as possible with their family.

Group 3 showed decreased food intake for 3 months prior to death which may have been due to dementia. It is estimated that about 81.5% of the residents in the facility experience dementia [36]. In patients with dementia, even if swallowing itself is not a major problem, as dementia progresses, the patient's perception of and interest in food declines, resulting in loss of appetite, poor food intake, dehydration, and malnutrition [37]. As dementia progresses, apraxia or inattention occurs, difficulty eating on one's own [38,39], dysphagia appears due to weakened gag reflex [40], and with advanced dementia it is estimated that 40.6% of nursing home residents have aspiration [41]. This study [41] reported the following in their study: a high proportion of nursing home residents with advanced dementia who died had complications such as pneumonia, febrile episodes, and eating problems during the last 3 months of life; and many of them received parenteral therapy, were hospitalized, taken to the emergency room, and underwent tube feeding during the last 3 months of life. By understanding this situation in advance, care providers can collaborate more effectively with medical providers, potentially leading to improved quality of end-of-life care.

Group 4 experienced decreased food intake, which gradually reduced for 6 months prior to death and may be due to oropharyngeal dysphagia owing to their frail condition. The prevalence of dysphagia among nursing home residents has been reported to range from 43.6% to 55.2% [42–44]. The prevalence of participantive dysphagia in care homes residents was 9% in another study [45]. However, nursing home residents did recognize their swallowing problems (participantive dysphagia) and considered them a natural symptom of their age or diseases. These conflicting data suggest that aspiration pneumonia in older adults residing in nursing homes is underdiagnosed or that dysphagia may be associated with a large proportion of pneumonia that is not thought to be caused by aspiration. The previous studies confirm the high prevalence rate of malnutrition in residents with participantive dysphagia [42,45,46]. Since malnutrition and oropharyngeal dysphagia is associated with mortality at 1 year [42,47], it is inferred that food intake is reduced 6 months before death. By understanding these changes, care providers can enhance the quality of end-of-life care.

In summary, the study interpreted the clinical trajectories of four distinct groups of changes in food intake before death among older residents in Japanese special nursing homes. In facilities such as special nursing homes, understanding and monitoring the daily changes in food intake recorded by care providers can lead to more appropriate end-of-life care. Moreover, understanding these clusters allows care providers to provide appropriate support in collaboration with medical providers such as nurses. The findings of this study suggest the possibility of such an approach.

This study has a few limitations. First, it was impossible to collect information on the diseases needed to classify and characterize the clusters of reduced food intake. We were, therefore, forced to speculate and discuss the status of each group of change in food intake that we classified. Future studies that include this information are required to determine the characteristics of each group of decreased food intake in detail and to examine prognostic indicators.

The second limitation is the generalizability of this study, which is due to the small number of participants. Future studies with larger and more diverse participants are necessary to confirm the external validity of the findings. Third, bias may have occurred due to the assignment of the missing data using the LOCF method. The missing data were substituted in the survey results because of missing food intake due to temporary hospitalization and return home. A cluster analysis performed by excluding participants with missing data yielded similar results. Fourth, we did not assess the inter-rater reliability of food intake. However, since they are in the same corporation, even though they are in multiple facilities, we assumed that the care provider's assessment of food intake is kept constant since it is entered using the same criteria. Further validation is needed. Fifth, although the five facilities in this study provide the same services as special nursing homes under the Japanese medical and long-term care systems, differences between the facilities emerged due to variations in resident characteristics. These differences may be influenced by the facility's philosophy, the allocation of long-term care providers and nurses, and the regional characteristics of their locations. However, the small number of participants in this study made it difficult to account for inter-facility factors in the analysis. In the future, research that considers differences between facilities will be necessary. However, the number of participants in this study was small, making it difficult to consider inter-facility factors in the analysis. Finally, the rate of food intake data was evaluated by care workers and nurses, and variations in inter-rater reliability may have affected data integrity. At the target facility, care workers and nurses hold monthly staff conferences to verify the appropriate rate of food intake assessment. The assessment of food intake at each meal is checked and evaluated not only by a care worker but also by several other staff members, including additional care workers and nurses. We believe that this system predominantly ensured accurate recording of the food intake data rate. In addition, as this study used average food intake data over one month, we believe this approach had a mitigating impact on inter-rater variability.

Despite these limitations, the study of indicators to classify clusters of decreased food intake based on data routinely collected in the special nursing home for older adults has the potential to improve the quality of end-of-life care provided in that setting for older adults.

## Conclusion

In conclusion, the food intake of older adults in the special nursing home for older adults changed during the last half-year before death, and cluster analysis classified them into the following four groups: immediate decrease; decrease from 1 month before death; decrease from 3 months before death; and gradual decrease for 6 months before death. The major contribution of this study is the conformation that eating decline is an important part of the end-of-life experience and changes in food intake should be expected and managed accordingly, with an understanding that these changes can occur in different clusters. We believe that by categorizing facility residents into several clusters of daily food intake, as done in this analysis, it will be possible to provide end-of-life care with a more appropriate approach for each individual.

## Supporting information

**S1 Appendix. Trajectory of changes in food intake in cluster 2, cluster 3, cluster 4, cluster 5 and cluster 6.**
(PDF)

## Acknowledgments

We want to thank all the facility's staff and residents who participated in the study.

## Author contributions

**Conceptualization:** Sakiko Fukui.

**Data curation:** Sakiko Fukui, Kasumi Ikuta.

**Formal analysis:** Sakiko Fukui, Kasumi Ikuta, Tatsuhiko Anzai, Kunihiko Takahashi.

**Funding acquisition:** Sakiko Fukui.

**Investigation:** Sakiko Fukui, Kasumi Ikuta.

**Methodology:** Sakiko Fukui, Kasumi Ikuta, Tatsuhiko Anzai, Kunihiko Takahashi.

**Project administration:** Sakiko Fukui.

**Supervision:** Tatsuhiko Anzai, Kunihiko Takahashi.

**Writing – original draft:** Sakiko Fukui.

**Writing – review & editing:** Kasumi Ikuta, Tatsuhiko Anzai, Kunihiko Takahashi.

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
