## [Decision Letter · Decision Letter 0]

4 Sep 2024

PONE-D-24-25291Classification of the trajectory of changes in food intake in special nursing home for oldest-old in the 6 months before death: a secondary analysisPLOS ONE

Dear Dr. Fukui,

Thank you for submitting your manuscript to PLOS ONE. After careful consideration, we feel that it has merit but does not fully meet PLOS ONE’s publication criteria as it currently stands. Therefore, we invite you to submit a revised version of the manuscript that addresses the points raised during the review process.

The manuscript is well evaluated by the reviewer.Respond to the reviewer's comments appropriately.

We look forward to receiving your revised manuscript.

Kind regards,

Masaki Mogi

Academic Editor

PLOS ONE

Journal Requirements:

1. When submitting your revision, we need you to address these additional requirements. Please ensure that your manuscript meets PLOS ONE's style requirements, including those for file naming. The PLOS ONE style templates can be found at https://journals.plos.org/plosone/s/file?id=wjVg/PLOSOne_formatting_sample_main_body.pdf and https://journals.plos.org/plosone/s/file?id=ba62/PLOSOne_formatting_sample_title_authors_affiliations.pdf 2. Thank you for stating the following financial disclosure: "This study was funded by Grant-in-Aid for Scientific Research B from Japan Society for the Promotion of Science  (18H03112)." Please state what role the funders took in the study.  If the funders had no role, please state: "The funders had no role in study design, data collection and analysis, decision to publish, or preparation of the manuscript." If this statement is not correct you must amend it as needed. Please include this amended Role of Funder statement in your cover letter; we will change the online submission form on your behalf. 3. In the online submission form, you indicated that "The datasets used and analyzed during the current study are available for further use.Corresponding author SF should be contacted if someone wants to request the data.Although public access to the database is closed, everyone can access the database of this study from the corresponding author on reasonable request without an administrative permissions to access the raw data.All data generated or analyzed during this study are included in this published article." All PLOS journals now require all data underlying the findings described in their manuscript to be freely available to other researchers, either 1. In a public repository, 2. Within the manuscript itself, or 3. Uploaded as supplementary information.This policy applies to all data except where public deposition would breach compliance with the protocol approved by your research ethics board. If your data cannot be made publicly available for ethical or legal reasons (e.g., public availability would compromise patient privacy), please explain your reasons on resubmission and your exemption request will be escalated for approval.

Reviewers' comments:

Reviewer's Responses to Questions

**Comments to the Author**

1. Is the manuscript technically sound, and do the data support the conclusions?

Reviewer #1: Partly

2. Has the statistical analysis been performed appropriately and rigorously? 

Reviewer #1: I Don't Know

3. Have the authors made all data underlying the findings in their manuscript fully available?

Reviewer #1: Yes

4. Is the manuscript presented in an intelligible fashion and written in standard English?

Reviewer #1: No

5. Review Comments to the Author

Reviewer #1: Thank you for the opportunity to review this manuscript on a very important topic, trajectories of food intake associated with end-of-life decline in institutionalized older adults.

General comments:

1. The manuscript is well-written but could benefit from a thorough review to fix grammar and spelling errors.

a. E.g., The “Settings” statement in the abstract; “revalently” in the first paragraph of the introduction. Other errors throughout

2. The claim that the findings can be used to predict timing to death is overstated and unsupported by the findings. This should be revised throughout the manuscript.

Methods:

1. Under Study Design, participants, and setting – the statement “we included 1281 older adults…” is misleading as this was not the included but total study population. Please clarify.

2. Data Collection: when was the level of care need reported? Presumably this would change for residents over time. It sounds like this was measured at one time point and used as a predictor of cluster membership?

3. Data Analysis: Please describe how you came up with 4 clusters? Was this data-driven or decided a priori?

4. I am unfamiliar with the statistical methods used to identify clusters. Suggest review by a statistical expert to assess the appropriateness of the specific method.

Results:

1. Missing data – this is mentioned in the methods and discussion but level of missing data is not described in the results. This would be helpful to fully assess the analysis.

2. The description of differences between groups (e.g., “group 1 was characterized by a longer length of stay (p=0.994)” is not supported by the analysis, and the authors should be careful not to make these comparative claims. There are some interesting general trends that could be commented on, but more appropriately in the discussion. The greatest difference was between facilities and this was not mentioned in the results.

Discussion:

1. Re: the statement “Despite these limitations, we believe that utilizing the food intake data recorded routinely in special nursing homes may assist care providers to estimate the approximate time until death and in providing appropriate end-of-life care...”

a. This claim is unsupported by your study. Since there is no control group of persons who did not die, and the observations are retrospective these findings cannot be considered predictors of time to death. Instead, I would suggest the recognition of various trajectories may prompt care providers to consider end-of-life decline and help with care management and expectations

2. The subsequent discussion of disease states and each of the trajectories is interesting, but highly speculative. The authors have mentioned this in the limitations, but should be more clearly stated prior to these paragraphs, for example, in the sentence “Therefore, we have provided an overview and discussion of each pattern of change regarding situations other than illness.” When discussing each trajectory, it would be helpful to reiterate that your claims are theoretical and not supported by the data. What types of information would be needed to support your theories?

3. A discussion of differences between facilities would be helpful, since the greatest differences in prevalence of these trajectories was found by facility. Do different facilities take in different level or profile of needs, for example?

4. Please identify the validity/reliability of staff-reported food intake, and note as a limitation.

Conclusion:

1. The conclusion over-emphasizes the prognostic value of the findings without being supported by the study because this was retrospective data collection. A care provider would not be able to identify which pattern is occurring “in the moment” nor would they be able to predict time to death based on changes in food intake alone. Rather, I would suggest the major contribution of this study is confirming that eating decline is an important part of the end-of-life experience and changes in food intake should be expected and managed accordingly, with an understanding that these changes can occur in different patterns.

6. PLOS authors have the option to publish the peer review history of their article (what does this mean? ). If published, this will include your full peer review and any attached files.

**Do you want your identity to be public for this peer review?** For information about this choice, including consent withdrawal, please see our Privacy Policy .

Reviewer #1: No

---

## [Author Response · Author response to Decision Letter 0]

1 Nov 2024

PONE-D-24-25291

Classification of the trajectory of changes in food intake in special nursing home for oldest-old in the 6 months before death: a secondary analysis

Journal Requirements:

Please ensure that your manuscript meets PLOS ONE's style requirements, including those for file naming. The PLOS ONE style templates can be found at https://journals.plos.org/plosone/s/file?id=wjVg/PLOSOne_formatting_sample_main_body.pdf

and https://journals.plos.org/plosone/s/file?id=ba62/PLOSOne_formatting_sample_title_authors_affiliations.pdf

Reply: We have prepared a resubmission manuscript in accordance with the Plos Ones’ Guidelines for Authors, and had the manuscript proofread by Editage, an English proofreading company.

2. Thank you for stating the following financial disclosure: "This study was funded by Grant-in-Aid for Scientific Research B from Japan Society for the Promotion of Science (18H03112)."

Reply: We have added the funding name and the roles of the funders to Acknowledgements in the manuscript as well as the cover letter, as per your suggestion.

3. In the online submission form, you indicated that "The datasets used and analyzed during the current study are available for further use. Corresponding author SF should be contacted if someone wants to request the data. Although public access to the database is closed, everyone can access the database of this study from the corresponding author on reasonable request without an administrative permissions to access the raw data.

All data generated or analyzed during this study are included in this published article."

Reply: We have uploaded the anonymized dataset used for the analysis in this study as an appendix file.

Reviewers' comments:

Reviewer's Responses to Questions

Comments to the Author

Reviewer #1

Thank you for the opportunity to review this manuscript on a very important topic, trajectories of food intake associated with end-of-life decline in institutionalized older adults.

General comments:

1. The manuscript is well-written but could benefit from a thorough review to fix grammar and spelling errors.

a. E.g., The “Settings” statement in the abstract; “revalently” in the first paragraph of the introduction. Other errors throughout

Reply: As indicated above, for this revision, we have used Editage for English proofreading.

2. The claim that the findings can be used to predict timing to death is overstated and unsupported by the findings. This should be revised throughout the manuscript.

Reply: Following your suggestion, we have revised the entire manuscript to ensure it is not overstated and unsupported. We have fully revised the manuscript, figures, and tables and all changes are noted in red.

Methods:

1. Under Study Design, participants, and setting – the statement “we included 1281 older adults…” is misleading as this was not the included but total study population. Please clarify.

Reply: Based on your comments, we consulted a statistical expert and performed a re-analysis. The statement n=1281 in the Abstract and Methods sections have been deleted and revised to n=69 after reanalysis.

Through this process, we consulted with a statistician to reconfirm the inclusion and exclusion criteria for the participants. We have clarified the exclusion criteria for the participants and have revised the following section of the Methods section (Page 4, Lines 127-130).

Additionally, due to the re-analysis, the number of participants has been changed from the previous n=70 to the present n=69. Although the number of subjects has changed, there were no significant changes in the cluster trajectories (Abstract and Page 4, Line 123).

In the Participants characteristics of the Results section, we also revised specific sentences (Page 6 Lines 202-204) and Figure 1 as follows.

2. Data Collection: when was the level of care need reported? Presumably this would change for residents over time. It sounds like this was measured at one time point and used as a predictor of cluster membership?

Reply: We added the sentence “As for the level of care needed, data were used from the earliest time point during the individual follow-up period of 6 months before death. (Page 5, Lines 153-155).

3. Data Analysis: Please describe how you came up with 4 clusters? Was this data-driven or decided a priori?

Reply: Four clusters were selected after reviewing the actual trajectories and participant distribution, ensuring clinical interpretability and utility. In this study, we considered clusters ranging from 2 to 6 groups. All cluster solutions have been included in the Appendix.

First, in Cluster 3, participants were categorized into “Decrease from 1 month before death (N=38)”, “Decrease from 3 months before death (N=7)”, and “Gradual decrease (N=24)”. In Cluster 4, participants were classified into “Immediate decrease (N=14)”, “Decrease from 1 month before death (N=24)”, “Decrease from 3 months before death (N=7)”, and “Gradual decrease (N=24)”. The group “Decrease from 1 month before death (N=38)” in Cluster 3 was divided into two groups in Cluster 4: “Immediate decrease (N=14)” and “Decrease from 1 month before death (N=24)”. Clinically, older adults whose food intake decreases immediately before death are more likely to experience sudden death compared to those whose food intake decreases one month before death. The approach to care, including interactions with the individual and their family, differs for those whose intake decreases one month before death. For these reasons, Cluster 4 was considered more appropriate than Cluster 3.

In Cluster 5, participants were categorized into “Immediate decrease (N=14)”, “Decrease from 1 month before death (N=24)”, “Decrease from 3 months before death (N=7)”, “Gradual decrease from 60% food intake (N=11)”, and “Gradual decrease from 40% food intake (N=13)”. This classification further divided the group “Gradual decrease (N=24)” from Cluster 4 into two groups, “Gradual decrease from 60% food intake (N=11)” and “Gradual decrease from 40% food intake (N=13)”. These two groups indicate a gradual decrease in food intake from a moderate level, but it is unlikely that this distinction would lead to significant differences in clinical care or interventions. Furthermore, excessive clustering can lead to overfitting and make practical application more difficult. Therefore, we selected Cluster 4 as we judged it to be the most appropriate both clinically and analytically.

Therefore, we have added the following information to the Methods and Results sections. Kindly review and confirm.

Method: Hierarchical cluster analysis, using Ward's method with squared Euclidean distance as the similarity measure [28,29], was conducted to identify clusters of clusters with high homogeneity within the clusters related to the cluster variable of change in the average per week dietary food intake from 6 months prior to death to classify clusters of change in food intake. Missing data were supplemented with the last observation carried forward (LOCF) method. The analyses were conducted with the number of clusters set at 2 to 6, respectively, and the final cluster model was selected based on the classification of the data and its clinical interpretability. (Page 6, Lines 173-180).

Result: The clusters appeared by the analysis with the number of clusters from 2 to 6 are shown in Appendix 1. We selected the number of clusters 4 as the final model based on clinical interpretability. The profiles of the 4 clusters, confirmed by linear mixed analysis considering covariates, are shown in Figure 2: Group 1 was named “Immediate decrease (n=14)”, Group 2 “Decrease from 1 month before death (n=24)”, Group 3 “Decrease from 3 months before death (n=7)”, and Group 4 “Gradual decrease for 6 months before death (n=24)”. (Page 7 Line 221-227).

4. I am unfamiliar with the statistical methods used to identify clusters. Suggest review by a statistical expert to assess the appropriateness of the specific method.

Reply: Thank you for your kind suggestion. We consulted with a statistician (Prof. Takahashi and Assoc Prof. Anzai) to re-examine the methods and results, as indicated above.

Results:

1. Missing data – this is mentioned in the methods and discussion but level of missing data is not described in the results. This would be helpful to fully assess the analysis.

Reply: Thank you for your suggestion. Missing data has been added to the results section and the table below also shows all missing data for each time point.

Results: Of the 25 time points (including the time of death: time 0) for 69 participants (total 1,725 data points), 27 time points (n=17) had missing data, where each had 1 to 2 missing time points. The most missing were observed at the time of death (time point 0) for 10 participants and at time point 1 for 9 participants. (Page 7 Lines 212-215).

2. The description of differences between groups (e.g., “group 1 was characterized by a longer length of stay (p=0.994)” is not supported by the analysis, and the authors should be careful not to make these comparative claims. There are some interesting general trends that could be commented on, but more appropriately in the discussion. The greatest difference was between facilities and this was not mentioned in the results.

Reply: Following your suggestion, I have carefully revised the description of differences between groups to avoid making unsupported comparative claims in the Results section (Page 7 Lines 228-232).

As for the difference between facilities, we have added information to the Limitations section as indicated below (Page 11, Lines 343-347).

Discussion:

1. Re: the statement “Despite these limitations, we believe that utilizing the food intake data recorded routinely in special nursing homes may assist care providers to estimate the approximate time until death and in providing appropriate end-of-life care...”

a. This claim is unsupported by your study. Since there is no control group of persons who did not die, and the observations are retrospective these findings cannot be considered predictors of time to death. Instead, I would suggest the recognition of various trajectories may prompt care providers to consider end-of-life decline and help with care management and expectations.

Reply: Thank you for your insightful comment. We have revised the sentence to "the recognition of various trajectories may prompt care providers to consider end-of-life decline and help with care management and expectations," as you suggested (Page 8 Lines 256-258).

2. The subsequent discussion of disease states and each of the trajectories is interesting, but highly speculative. The authors have mentioned this in the limitations, but should be more clearly stated prior to these paragraphs, for example, in the sentence “Therefore, we have provided an overview and discussion of each pattern of change regarding situations other than illness.” When discussing each trajectory, it would be helpful to reiterate that your claims are theoretical and not supported by the data. What types of information would be needed to support your theories?

Reply: We have revised the manuscript to avoid speculative expressions in the section you pointed out. Additionally, we have included the suggested sentence, "Therefore, we have provided an overview and discussion of each pattern of change regarding situations other than illness (Page 8 Lines 258-259)." Furthermore, as per your suggestion, we reiterated that our claims are theoretical and not supported by the data. We also revised 4 references (present manuscript No. 8-11) to support these theories.

3. A discussion of differences between facilities would be helpful, since the greatest differences in prevalence of these trajectories was found by facility. Do different facilities take in different level or profile of needs, for example?

Reply: Although the five facilities in this study provide the same services as special nursing homes under the Japanese medical and long-term care systems, differences between the facilities emerged due to variations in resident characteristics. These differences may be influenced by the facility's philosophy, the allocation of long-term care staff and nurses, and the regional characteristics of their locations. However, the small number of subjects in this study made it difficult to account for inter-facility factors in the analysis. In the future, research that considers differences between facilities will be necessary.

The above has been listed as a Limitation (Page 11, Lines 343-347).

4. Please identify the validity/reliability of staff-reported food intake, and note as a limitation.

Reply: We have designed the study to identify the validity and reliability of staff-reported food intake as much as possible. We have added the description of our previous paper (Ikuta, 2024), which presents the data of this study, to this manuscript as well.

Methods: The assessment of food intake at each meal undergoes an inter-rater evaluation process by multiple staff members, and monthly conferences are held to validate the evaluation of food intake data (Page 5, Lines 155-157).

Limitation: Finally, the rate of food intake data was evaluated by care workers and nurses, and variations in inter-rater reliability may have affected data integrity. At the target facility, care workers and nurses hold monthly staff conferences to verify the appropriate rate of food intake assessment. The assessment of food intake at each meal is checked and evaluated not only by a care worker but also by several other staff members, including additional care workers and nurses. We believe that this system predominantly ensured accurate recording of the food intake data rate. In addition, as this study used average food intake data over one month, we believe this approach had a mitigating impact on inter-rater variability (Page 11, Line 349-357).

Conclusion:

1. The conclusion over-emphasizes the prognostic value of the findings without being supported by the study because this was retrospective data collection. A care provider would not be able to identify which pattern is occurring “in the moment” nor would they be able to predict time to death based on changes in food intake alone. Rather, I would suggest the major contribution of this study is confirming that eating decline is an important part of the end-of-life experience and changes in food intake should be expected and managed accordingly, with an understanding that these changes can occur in different patterns.

Reply: Thank you for your advice. We have removed the overemphasized expressions regarding prognosis and added the suggested sentence, "the major contribution of this study is the conformation that eating decline is an important part of the end-of-life experience and changes in food intake should be expected and managed accordingly, with an understanding that these changes can occur in different patterns," to the manuscript (Page 11 Lines 368-371).

---

## [Editor Report · Decision Letter 1]

6 Feb 2025

Classification of the trajectory of changes in food intake in special nursing home for oldest-old in the 6 months before death: a secondary analysis

PONE-D-24-25291R1

Dear Dr. Fukui,

We’re pleased to inform you that your manuscript has been judged scientifically suitable for publication and will be formally accepted for publication once it meets all outstanding technical requirements.

Kind regards,

Masaki Mogi

Academic Editor

PLOS ONE

Additional Editor Comments (optional):

The manuscript has been well revised and improved according to the reviewer’s suggestions. No further comment.
---

## [Editor Report · Acceptance letter]

PONE-D-24-25291R1

PLOS ONE

Dear Dr. Fukui,

I'm pleased to inform you that your manuscript has been deemed suitable for publication in PLOS ONE. Congratulations! Your manuscript is now being handed over to our production team.

Kind regards,

on behalf of

Dr. Masaki Mogi

Academic Editor

PLOS ONE